# Relationship quality and mental health during COVID-19 lockdown

Christoph Pieh[1]*, Teresa O´Rourke[1], Sanja Budimir[1,2], Thomas Probst[1]

**1** Department for Psychotherapy and Biopsychosocial Health, Danube University Krems, Krems an der Donau, Austria, **2** Department of Work, Organization and Society, Ghent University, Ghent, Belgium

* Christoph.pieh@donau-uni.ac.at

## Abstract

Catastrophes are known to have an impact on relationships as well as on mental health. This study evaluated differences in several mental health and well-being measures according to relationship quality during the Coronavirus Disease (COVID-19) pandemic and related lockdown measures. A cross-sectional online survey was launched four weeks after lockdown measures were implemented in Austria. Relationship quality was measured with the Quality of Marriage Index (QMI), and mental health measures included quality of life (WHO-QOL BREF psychological domain), well-being (WHO-5), depression (PHQ-9), anxiety (GAD-7), stress (PSS-10), and sleep quality (ISI). ANOVAs with Bonferroni-corrected post-hoc tests and Chisquared tests were applied. In all mental health scales, individuals with good relationship quality (n = 543) scored better than individuals with poor relationship quality (n = 190) or without relationship (n = 272). The odds ratios (OR) between the poor and good relationship quality groups were 3.5 for the PHQ-9, 3.4 for the GAD-7, and 2.0 for the ISI. Additionally, individuals without no relationship scored better on all scales than individuals with poor relationship quality (all p-values < .05). Relationship quality was related to mental health during COVID-19. The prevalence of depressive symptoms increased according to relationship quality from 13% up to 35%. Relationship per se was not associated with better mental health, but the quality of the relationship was essential. Compared to no relationship, a good relationship quality was a protective factor whereas a poor relationship quality was a risk factor.

## Introduction

As the Coronavirus disease 2019 (COVID-19) has spread quickly throughout the world [1], most governments have implemented restrictions to prevent the uncontrolled spreading of the virus. Although social distancing and other measures such as the use of personal protective equipment could help to contain the uncontrolled spreading of SARS-CoV-2 [1], they seem to negatively affect mental health [2].

Associations between mental health and relationship quality have been found in several previous studies [3, 4]. There is a considerable amount of evidence showing that married individuals enjoy better mental health than never-married and previous married individuals [5].

**Data Availability Statement:** All relevant data are within the manuscript and its Supporting Information files.

**Funding:** The author(s) received no specific funding for this work.

**Competing interests:** The authors have declared that no competing interests exist.

In times of COVID-19, a survey from India showed that married participants had 40% lower odds of developing anxiety during COVID-19 lockdown than unmarried participants [6]. Yet, the following examples show that the relationship between marriage / relationship and mental health seems to be moderated by marriage / relationship quality [7]. Being married per se is not universally beneficial, rather, the satisfaction and support associated with such a relationship is important [3]. For example, results from Frech and Williams [8] suggest that the effect of marriage on depression is dependent on the quality of the marital relationship. Furthermore, single people have better mental health outcomes than people who are unhappily married [3]. Findings from a population-based study in the US showed that relationship discord can be associated with higher risks for mood and anxiety disorders [9]. These results go in line with a population-based survey in Australia showing that a better relationship quality is associated with less depression and anxiety symptoms than worse relationship quality [10]. In addition, lack of quality of social relationships was found to be a major risk factor for major depression [4]. Viceverse, high marital quality was associated with lower stress and depression, but also with lower blood pressure as well as higher slow-wave sleep [3].

Although several assumptions about an increase of divorce rates due to COVID-19 pandemic and related lockdown measures have been made on the news, it might be too early to assess the impact on divorce rates yet. However, as known from former catastrophes, such as the hurricane Hugo, such challenging times can have an impact on relationship, marriage, birth, and divorce rates [11]. Divorce rates increased in the affected compared to unaffected counties [11]. However, following the attacks from September 11[th], 2001 in New York City, divorce rates decreased [12]. Maybe there is an opposing effect if the disaster is manmade or not.

Due to the COVID-19 pandemic most governments implemented quarantine measures. In Austria, COVID-19 social distancing measures became obligatory on 16th of March 2020. Only in some exceptions it was allowed leaving the own household. This constitutes an extraordinary situation, not only for individually mental health, but also for relationships. The aim of the current study was to evaluate the effect of relationship quality on mental health and well-being indicators in a representative population sample in Austria during COVID-19 lockdown.

## Methods

### Study design

A cross-sectional online survey was performed in Austria using the Qualtrics® population survey platform. Qualtrics is an experience management company with a platform for online surveys and participants recruitment, available at https://www.qualtrics.com. Apart from hosting the online survey, Qualtrics provides organization and collection of data based on predefined sample, methodology, design, and qualifying question syntax provided by researchers. It also offers quality check including attention fillers, survey timings as well as replacement of unusable data.

A representative sample with a minimum sample size of 1,000 according to age, gender, education, and region was a specified a priori. Qualtrics® provided us with the final sample of N = 1,005 participants. The survey was launched four-weeks after quarantine measures were implemented in Austria. Participants were contacted by the Qualtrics project team who organized and coordinated data collection. As part of the scoping process, Qualtrics implemented age, gender, educational, and regional quotas based on Austrian population census data. Overall, the target sample was attained within ten days, after which the survey closed. COVID-19 lockdown was officially implemented in the Austria on 16[th] of March 2020, and the survey

started on 10<sup>th</sup> of April 2020 for 10 days. As all used questionnaires relate to the last two or four weeks, we started the survey four weeks after lockdown.

## Governmental restrictions during the survey

COVID-19 social distancing measures became obligatory on 16th of March 2020 I Austria (COVID- 19 lockdown). To summarize; entering public places was strictly prohibited and only allowed in some exceptions There were only the following five exceptions of the ban to enter public places. Activities to avert an immediate danger to life, limb, or property; professional activity (if home-office is not possible); errands to cover necessary basic needs; care and assistance for people in need of support; exercise outdoors (e.g. running, walking) alone and with pets / people living in the same household. A distance of at least 1 meter to other people has to be ensured.

## Questionnaires

All used questionnaires are validated in German language and were presented in a forced choice answer format. Thus, there are no missing items in the data set.

**Relationship satisfaction.** The Quality of Marriage Index (QMI) is a 6-item internationally widely-used instrument assessing relationship quality. The German QMI demonstrates good item characteristics and excellent reliability ($\alpha$ = .94), adequate psychometric properties and reliably measures relationship quality across gender and age [13]. The recommended cut-off score for the German version is 34 for male and women with a sensitivity of 88%, specificity of 85%, and Youden-Index $J$ = .73 [13].

**Quality of life.** The WHOQOL-BREF is a 26 items self-rating questionnaire, which measures physical health, psychological health, social relationships, and environment during the last two weeks. It allows a reliable, valid, and brief assessment of quality-of-life. To indicate the psychological aspect of quality of life in the present study, the psychological domain (6 items) was used. The WHOQOL-BREF psychological domain norm for the general population has been reported to be 70.6 (14.0) [14].

**Well-being.** The WHO-5 Well Being Index was used to measure well-being within the past two weeks. It consists of five self-rating items on six-point Likert scales with a raw score range from 0 (absence of well-being) to 25 (maximal well-being), whereas a higher score is indicative of better well-being. Reliability and validity of the WHO-5 have been well established [15].

**Perceived stress.** The 10-item perceived stress scale (PSS-10) was used to measure stress severity during the last month [16]. The items are scored on a Likert scale from 0 to 4, with higher scores indicating higher stress severity. The PSS-10 is a reliable and valid tool to measure stress severity.

**Depressive symptoms.** To measure depressive symptoms, the depression module of the Patient Health Questionnaire (PHQ-9) was used, which constitutes a validated screening tool for depression [17]. The 9 self-rating items are scored on a four-point scale from 0 to 3, with a total severity score ranging from 0 to 27. The clinical cut-off points are 5 for mild depression, 10 for moderate depression and 15 or higher for moderate to severe depression. To define clinically relevant depression, the 10point cut-off score was used in the present study.

**Anxiety symptoms.** The Generalized Anxiety Disorder 7 scale (GAD-7) was used to measure anxiety symptoms [18]. This validated screening tool for anxiety consists of 7 self-rating items scored on a four point scale, from 0 to 3. The total anxiety severity score therefore ranges from 0 to 21. The clinical cut-off points are set at 5 for mild, 10 for moderate and 15 for severe anxiety symptoms.

Clinically relevant anxiety was defined with the 10-point cut-off score in the current study.

**Sleep quality.** Sleep quality was measured with the Insomnia Severity Index (ISI), which is a validated 7-item self-report on sleep quality and insomnia [19]. The items are scored from 0 to 4 on a five-point scale. Symptom severity categories are: no clinically significant insomnia (0–7 points), subthreshold insomnia (8–14 points), clinical insomnia (moderate severity) (15–21 points), and clinical insomnia (severe) (22–28 points). To define moderate (i.e. clinically relevant) insomnia, the cut-off score of $\geq 15$ was used in this study.

### Study sample

All N = 1,005 participants were analyzed. The sample was specified a priori with a minimum of 1000 participants according to age, gender, education, and region. Qualtrics provided us with the final sample of N = 1005. All of these participants were analyzed.

### Statistical analysis

All data were analyzed using SPSS version 24. Descriptive statistics were computed for the demographic characteristics and mental health scales. Based on the literature, we applied the cutoff $\geq 10$ to examine the proportion of cases with clinically relevant depression (PHQ-9), anxiety (GAD-7), and $\geq 15$ for insomnia symptoms.

ANOVAs and Bonferroni-corrected post-hoc tests were calculated to evaluate differences in mental health indicators (depression, anxiety, stress, well-being, sleep quality, quality of life), comparing the following three groups: good relationship quality, poor relationship quality and no relationship. For ANOVAs, $\eta^2$ was used as effect size, which can be interpreted as follows: small ($\eta^2$ = .01 to .06), medium ($\eta^2$ = .06 to .14), and large ($\eta^2 \geq$ .14). Moreover, t-tests for independent samples were conducted to 1) compare the QMI scores of our study with the QMI scores provided by Zimmermann et al. [13] and 2) to compare the QMI scores for those coded as having good relationship quality vs. those coded as having poor relationship quality. P-values <0.05 were considered statistically significant (2-sided tests).

### Ethical considerations

This study was approved by the Ethics Committee of the Danube University Krems and conducted in accordance with the Declaration of Helsinki. All participants gave electronic informed consent for participation and before completing the questionnaires and received an expense allowance from Qualtrics. Data was collected anonymously without IP addresses or GPS tracking, and this procedure was approved by the data protection officer of the Danube-University Krems, Austria.

## Results

The mean QMI score was M = 36.95 (SD = 9.11) for all n = 733 individuals being in a relationship. Characteristics of this sample including age, gender, education, income, region, child care, and living situation are presented in Table 1.

Based on the 34-point QMI cut-off for the group with poor relationship quality (n = 190), the mean was M = 24.15 (SD = 8.08). For the group with good relationship quality (n = 543), the mean was M = 41.43 (SD = 3.42). Comparisons between the three relationship groups (good vs. poor relationship quality as well as no relationship as control group) regarding age and gender are presented in Table 2.

All mental health indicators (depression, anxiety, stress, well-being, sleep quality, quality of life) were significantly different between the three relationship groups (Table 3).

**Table 1. Sample description for participants living in a relationship (n = 733).**

|  | *n* (%) |
|---|---|
| **Age** |  |
| 18–24 | 63 (8.6) |
| 25–34 | 132 (18.0) |
| 35–44 | 145 (19.8) |
| 45–54 | 160 (21.8) |
| 55–64 | 135 (18.4) |
| 65+ | 98 (13.4) |
| **Gender** |  |
| Male | 367 (50.1) |
| Female | 366 (49.9) |
| **Region** |  |
| Burgenland | 23 (3.1) |
| Lower Austria | 150 (20.5) |
| Vienna | 155 (21.1) |
| Carinthia | 49 (6.7) |
| Styria | 101 (13.8) |
| Upper Austria | 122 (16.6) |
| Salzburg | 51 (7.0) |
| Tyrol | 50 (6.8) |
| Vorarlberg | 32 (4.4) |
| **Highest level of education** |  |
| Less than high school | 1 (0.1) |
| Lower secondary education | 18 (2.5) |
| Vocational training (Apprenticeship) | 237 (32.3) |
| A-levels | 207 (28.2) |
| Tertiary education (College. University) | 270 (36.8) |
| **Living situation** |  |
| Apartment | 140 (19.1) |
| Apartment with balcony or terrace | 252 (34.4) |
| House with or without garden | 341 (46.5) |
| **Childcare** |  |
| No child(ren) in need of care | 516 (70.4) |
| Care for child(ren) alone | 33 (4.5) |
| Shared childcare | 169 (23.1) |
| Partner cares for child | 15 (2) |
| **Job situation** |  |
| No job (did not have on before) | 119 (16.2) |
| No job (had one before) | 64 (8.7) |
| Home Office | 207 (28.2) |
| Job at the same workplace (not home office) | 145 (19.8) |
| Reduced working hours | 73 (10.0) |
| Retired | 125 (17.1) |
| **Monthly household net income** |  |
| < € 1.000,- | 22 (3.0) |
| € 1.000,- to € 2.000,- | 125 (17.1) |
| € 2.000,- to € 3.000,- | 236 (32.2) |
| € 3.000,- to € 4.000,- | 175 (239) |

(*Continued*)

**Table 1.** (Continued)

|  |  | *n* (%) |
|---|---|---|
|  | > € 4.000,- | 175 (23.9) |
| **Living arrangement** |  |  |
|  | Living alone | 63 (8.6) |
|  | Living separately | 11 (1.5) |
|  | Married | 403 (55.0) |
|  | Divorced | 13 (1.8) |
|  | Living with partner | 239 (32.6) |
|  | Widowed | 4 (.5) |

Bonferroni-corrected post-hoc tests (Table 4) performed to follow-up the significant ANO-VAs revealed that–in all scales–individuals with a good relationship quality had significantly better scores compared to individuals with a poor relationship quality as well as compared to individuals without relationship (all $p < .05$). In addition, individuals without relationship had better scores–again in all scales–than individuals with a poor relationship quality (all $p < .05$) (Table 4).

The odds ratios (OR) between the poor and good relationship quality groups were 3.5 [CI: 2.4, 5.2] (PHQ-9), 3.4 [CI: 2.3, 5.0] for the GAD-7, and 2.0 [1.3, 3.0] for the ISI.

## Discussion

This study examined the relationship status as well as relationship quality on a broad range of mental health and well-being indicators during COVID-19 lockdown. We found clinically relevant differences according to relationship quality as well as to relationship status throughout all tested scales. Individuals with good relationship quality showed better mental health than individuals with poor relationship quality or no relationship. Furthermore, individuals with poor relationship quality performed significantly worse in all mental health scales.

The mean QMI score in our sample of $M = 36.95$ (SD = 9.11) was slightly, but significant lower compared to the data of a study from 2019 with $M = 39.05$ (SD = 6.43), which was performed at a population sample from Germany ($t(1115.24) = -5.58$; $p < .001$) [13]. It could be that relationship quality suffered during COVID-19 or that the sample recruited in Germany differs from our sample in confounders.

The findings with regard to good mental health in individuals with good relationship quality is in line with previous research. According to a review on marital quality and depression,

**Table 2. Comparisons between the three relationship groups regarding age and gender.**

|  |  | Relationship Groups | | | Total | Statistic |
|---|---|---|---|---|---|---|
|  |  | **Good relationship quality** | **Poor relationship quality** | **No relationship** |  |  |
| **Age n (%)** | 18–24 | 53 (9.8) | 10 (5.3) | 55 (20.2) | 118 (11.7) | $\chi^2(10) = 36.67$; $p < .001$ |
|  | 25–34 | 100 (18.4) | 32 (16.5) | 34 (12.5) | 166 (16.5) |  |
|  | 35–44 | 98 (18.0) | 47 (24.7) | 40 (14.7) | 185 (18.4) |  |
|  | 45–54 | 119 (21.9) | 41 (21.6) | 62 (22.8) | 222 (22.1) |  |
|  | 55–64 | 97 (17.9) | 38 (20.0) | 46 (16.9) | 181 (18.0) |  |
|  | 65+ | 76 (14.0) | 22 (11.6) | 35 (12.9) | 133 (13.2) |  |
| **Gender n (%)** | Male | 274 (50.5) | 93 (48.9) | 108 (39.7) | 475 (47.3) | $\chi^2(2) = 8.68$ $p = .013$ |
|  | Female | 269 (49.5) | 97 (51.1) | 164 (60.3) | 530 (52.7) |  |
|  | **Total** | **543 (100)** | **190 (100)** | **272 (100)** | **1005 (100)** |  |

**Table 3. Results for depression, anxiety, insomnia, psychological quality of life, well-being, and perceived stress between relationship groups.**

| | | Good relationship quality | Poor relationship quality | No relationship | Total | Statistic |
|---|---|---|---|---|---|---|
| **PHQ-9 n (%)** | <10 | 470 (86.6) | 123 (64.7) | 201 (73.9) | 794 (79.0) | $\chi^2(1) = 46.26$; $p < .001$ |
| | >= 10 | 73 (13.4) | 67 (35.3) | 71 (26.1) | 211 (21.0) | |
| **GAD-7 n (%)** | <10 | 476 (87.7) | 129 (67.9) | 209 (76.8%) | 814 (81.0) | $\chi^2(1) = 39.91$; $p < .001$ |
| | >= 10 | 67 (12.3) | 61 (32.1) | 63 (23.2) | 191 (19.0) | |
| **ISI n (%)** | <15 | 474 (87.3) | 148 (77.9) | 225 (82.7) | 847 (84.3) | $\chi^2(1) = 10.07$; $p = .007$ |
| | >= 15 | 69 (12.7) | 42 (22.1) | 47 (17.3) | 158 (15.7) | |
| | Total | 543 (100) | 190 (100) | 272 (100) | 1005 (100) | |
| **PHQ-9** | M | 4.87 | 8.41 | 7.25 | 6.19 | $F(2,1004) = 40.37$; $p < .001$; $\eta^2 = .074$ |
| | SD | 4.78 | 5.40 | 5.83 | 5.40 | |
| **GAD-7** | M | 4.91 | 7.86 | 6.28 | 5.84 | $F(2, 1004) = 31.32$; $p < .001$; $\eta^2 = .058$ |
| | SD | 4.29 | 4.77 | 4.92 | 4.70 | |
| **ISI** | M | 7.46 | 10.17 | 8.69 | 8.31 | $F(2, 1004) = 17.21$; $p < .001$; $\eta^2 = .033$ |
| | SD | 5.42 | 5.98 | 5.74 | 5.70 | |
| **WHOQOL BREF psychological domain** | M | 75.43 | 60.16 | 65.40 | 69.83 | $F(2,1004) = 64.66$; $p < .001$; $\eta^2 = .114$ |
| | SD | 16.01 | 18.34 | 20.02 | 18.70 | |
| **WHO-5** | M | 16.42 | 12.35 | 14.20 | 15.05 | $F(2,1004) = 48.68$; $p < .001$; $\eta^2 = .088$ |
| | SD | 4.81 | 5.25 | 5.76 | 5.40 | |
| **PSS-10** | M | 14.28 | 19.12 | 17.15 | 15.97 | $F(2,1004) = 36.64$; $p < .001$; $\eta^2 = .068$ |
| | SD | 6.91 | 7.13 | 7.85 | 7.47 | |

p: p-values (2-tailed); n: frequencies; M: mean score; SD: standard deviation, $\chi^2$: Chi-square; ISI: Insomnia Severity Index, GAD-7 (Generalized Anxiety Disorder 7 scale); PHQ9: Patient Health Questionnaire 9 scale; PSS-10: Perceived Stress Scale 10; WHO-5: Well-being questionnaire of the World Health Organization (WHO); WHO-QOL BREF: Quality of Life questionnaire of the World Health Organization (WHO).

numerous cross-sectional and longitudinal studies provide evidence for an association between marital dissatisfaction and depressive symptoms in younger and middle aged adults, as well as older adults [20]. Some findings of this review also suggest that poor marital quality is associated with higher depression relapse rates. Our result, that people with poor relationship quality showed the poorest mental health, even compared to people without relationships, is in contrast to the population-based study of Leach, Butterworth, Olesen, and Mackinnon [10], who reported that persons with poor relationship quality and singles had similar depression and anxiety scores. However, single individuals had better mental health outcomes than people who were unhappily married in another study [3], which corresponds to our results. The odds ratio for depression was 3.5 meaning a higher risk for individuals with poor relationship quality compared to individuals with good relationship quality, OR was 3.4 for anxiety symptoms, and 2.0 for clinical insomnia. The OR for depression is higher than the one reported in a previous study (OR 2.60) for depression [4]. However, as Teo and colleagues [4] measured overall relationship quality across different areas (spouse or partner, family members, and friends) with a self-constructed eight-item scale., a comparison is only possible to a limited extent.

The following limitations have to be considered, when interpreting the results: We performed a cross-sectional study, which allows no causal conclusions. A second measurement point before the COVID-19 lockdown would be necessary to draw causal conclusions. Therefore, we cannot say whether relationship quality had an impact on mental health or whether mental health influenced relationship quality or both. Although the sample is representative for age, gender, education, and region, it is not representative for combinations of these variables, e.g. age interlocked with gender. The generalizability is questionable due to a rather

**Table 4. Results for Bonferroni-corrected post-hoc tests.**

| | | | Mean difference (I-J) | SE | p | 95% CI |
|---|---|---|---|---|---|---|
| **PHQ-9** | | | | | | |
| | Good relationship quality | Poor relationship quality | -3.54 | .438 | < .001 | [-4.59; -2.49] |
| | | No relationship | -2.38 | .3876 | < .001 | [-3.30; -1.45] |
| | Poor relationship quality | Good relationship quality | 3.54 | .438 | < .001 | [2.49; 4.59] |
| | | No relationship | 1.16 | .492 | .055 | [-.02; 2.34] |
| **GAD-7** | | | | | | |
| | Good relationship quality | Poor relationship quality | -2.96 | .384 | < .001 | [-3.88; -2.03] |
| | | No relationship | -1.38 | .339 | < .001 | [-2.19; -.56] |
| | Poor relationship quality | Good relationship quality | 2.96 | .384 | < .001 | [2.03; 3.88] |
| | | No relationship | 1.58 | .431 | .001 | [.55; 2.61] |
| **ISI** | | | | | | |
| | Good relationship quality | Poor relationship quality | -2.70 | .473 | < .001 | [-3.84; -1.57] |
| | | No relationship | -1.23 | .417 | .010 | [-2.23; -.23] |
| | Poor relationship quality | Good relationship quality | 2.70 | .473 | < .001 | [1.57; 3.84] |
| | | No relationship | 1.47 | .531 | .017 | [.20; 2.75] |
| **PSS-10** | | | | | | |
| | Good relationship quality | Poor relationship quality | -4.84 | .608 | < .001 | [-6.30; -3.38] |
| | | No relationship | -2.87 | .536 | < .001 | [-4.15; -1.58] |
| | Poor relationship quality | Good relationship quality | 4.84 | .608 | < .001 | [3.38; 6.30] |
| | | No relationship | 1.97 | .682 | .012 | [.34; 3.61] |
| **WHO-5** | | | | | | |
| | Good relationship quality | Poor relationship quality | 4.07 | .435 | < .001 | [3.02; 5.11] |
| | | No relationship | 2.22 | .384 | < .001 | [1.30; 3.14] |
| | Poor relationship quality | Good relationship quality | -4.07 | .434 | < .001 | [-5.11; -3.02] |
| | | No relationship | -1.85 | .488 | < .001 | [-3.02; -.68] |
| **WHO-QOL BREF psychological domain** | | | | | | |
| | Good relationship quality | Poor relationship quality | 15.27 | 1.49 | < .001 | [11.71; 18.83] |
| | | No relationship | 10.03 | 1.31 | < .001 | [6.89; 13.17] |
| | Poor relationship quality | Good relationship quality | -15.27 | 1.49 | < .001 | [-18.83; -11.71] |
| | | No relationship | -5.24 | 1.67 | .005 | [-9.23; -1.24] |

p: p-values (2-tailed); n: frequencies; M: mean score; SD: standard deviation, $\chi^2$: Chi-square; ISI: Insomnia Severity Index, GAD-7 (Generalized Anxiety Disorder 7 scale); PHQ9: Patient Health Questionnaire 9 scale; PSS-10: Perceived Stress Scale 10; WHO-5: Well-being questionnaire of the World Health Organization (WHO); WHO-QOL BREF: Quality of Life questionnaire of the World Health Organization (WHO).

small sample size. Furthermore, only self-rating scales were used to assess mental health indicators (depression, anxiety, stress, well-being, sleep quality, quality of life) without an additional clinical interview or assessment. Thus, it makes the interpretation of the results vague. Especially, as screening questionnaires can overestimate the prevalence for e.g. depression, as reported by Thombs et al. [21]. Thus, in our sample the prevalence of participants scoring above the recommended cut-offs scores might be too high. The current results were compared to previous studies, which were conducted earlier and in other countries. We used the recommended cut-off score of the German version (34 points) of the QMI. However, the original U.S. questionnaire from 1,976 recommended a different cut-off score (29 points) [22, 23]. Still, by using the cut-off score of 29 points in our study we found similar effects. The number of participants varied in the three compared groups, with the subsample of good relationship quality being twice as high as the other subsamples. Another drawback is the missing

information on response rates. Due to the forced choice answer format, it is possible that participants dropped out of the questionnaire. Unfortunately, we do not know how many participants were contacted and declined to participate or started and stopped filling out the questionnaire at some point. Furthermore, no clear inclusion or exclusion criteria was formulated when recruiting the representative sample. The duration of four weeks may also be short to make informed statements about psychological effects, as symptoms might occur delayed.

In sum, the lockdown is a challenge especially for those with poor relationship quality. Those with poor relationship quality scored worst in all measures and showed almost three times higher risk for depressive symptoms (12% vs. 35%) as well as for anxiety symptoms (12% vs. 32%). As the individuals with good relationship quality scored best on the mental health scales and those without relationship between the ones with good and poor relationship quality. It underlines the fact that not only but especially in times like this, the choice of partner should be carefully considered.

## Supporting information

**S1 Data.**
(DOCX)

## Author Contributions

**Conceptualization:** Christoph Pieh, Sanja Budimir, Thomas Probst.

**Data curation:** Teresa O´Rourke, Thomas Probst.

**Formal analysis:** Teresa O´Rourke, Sanja Budimir, Thomas Probst.

**Methodology:** Sanja Budimir.

**Project administration:** Christoph Pieh.

**Software:** Teresa O´Rourke, Sanja Budimir.

**Writing – original draft:** Christoph Pieh.

**Writing – review & editing:** Christoph Pieh, Teresa O´Rourke, Sanja Budimir, Thomas Probst.

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
