## [Decision Letter · Decision Letter 0]

20 Jul 2020

PONE-D-20-15368

Relationship quality and mental health during COVID-19 lockdown in Austria

PLOS ONE

Dear Dr. Pieh,

Thank you for submitting your manuscript to PLOS ONE. After careful consideration, we feel that it has merit but does not fully meet PLOS ONE’s publication criteria as it currently stands. Therefore, we invite you to submit a revised version of the manuscript that addresses the points raised during the review process.

We look forward to receiving your revised manuscript.

Kind regards,

Ali Montazeri

Academic Editor

PLOS ONE

Journal Requirements:

2. Please amend either the title on the online submission form (via Edit Submission) or the title in the manuscript so that they are identical.

Reviewers' comments:

Reviewer's Responses to Questions

**Comments to the Author**

1. Is the manuscript technically sound, and do the data support the conclusions?

Reviewer #1: Partly

Reviewer #2: Partly

2. Has the statistical analysis been performed appropriately and rigorously? 

Reviewer #1: N/A

Reviewer #2: Yes

3. Have the authors made all data underlying the findings in their manuscript fully available?

Reviewer #1: No

Reviewer #2: Yes

4. Is the manuscript presented in an intelligible fashion and written in standard English?

Reviewer #1: Yes

Reviewer #2: No

5. Review Comments to the Author

Reviewer #1: - What is the  justification for the study sample size?

- Is four weeks enough for observing mental health events? What is justification for choosing this time period?

- Please provide sufficient explanations regarding the Austrian versions of the tools used.

- It is necessary to accurately assess the value of statistical tests on the relationship between variables by a statistician.

Reviewer #2: PONE-D-20-15368

The manuscript entitled ‘Relationship quality and mental health during COVID-19 lockdown in Austria’ aimed to evaluate the effect of relationship quality on mental health and well-being indicators in Austria during COVID-19 lockdown. The methodology needs improvements. I have provided some comments as follows:

- Were there eligibility criteria including exclusion or inclusion criteria?

- No match between the sample size mentioned in study sample (1000) and the result (1009).

- The sample of your study are people who were in quarantine, and they were asked to fill out 7 questionnaires, what was the response rate? All 1009 completed the questionnaires without missing one? Is this ethical to administer 7 questionnaires?

- What was your definition of mental health indicators?

- Please don’t report results in the method like study sample subheading!

- Please don’t compare the result of your study with another as you mentioned in result line 6.

- Please don’t re-mention the results in discussion.

6. PLOS authors have the option to publish the peer review history of their article (what does this mean?). If published, this will include your full peer review and any attached files.

Reviewer #1: No

Reviewer #2: No

---

## [Author Response · Author response to Decision Letter 0]

11 Aug 2020

Response to reviewer comments

Reviewer #1:

 What is the justification for the study sample size?

• To obtain a representative population sample according to age, gender, education, and region we specified the sample size a priori with a minimum of 1000 participants. Qualtrics then provided us with the final sample of N=1005 participants. We clarified the consideration for this decision within the manuscript. 

Is four weeks enough for observing mental health events? What is justification for choosing this time period?

• We chose this time period, because all the used scales relate to the last two or four weeks. However, mental health events can occur delayed and we can´t make a statement about it. We added this limitation in the manuscript. As we observed significantly effects on mental health, the investigation does not appear to have been carried out too early. 

Please provide sufficient explanations regarding the Austrian versions of the tools used.

• All used questionnaires (WHO-QOL BREF, WHO-5, PHQ-9, GAD-7, PSS-10, and ISI) are validated in German language and references are provided. 

It is necessary to accurately assess the value of statistical tests on the relationship between variables by a statistician.

• Thanks for pointing this out. We have made statistical considerations and decided to calculate t-tests and variance analysis to analyze group differences instead of other statistical possibilities (e.g. regression analysis), because these methods are robust against violations of the respective requirements with large samples. 

• Thanks for your feedback to our manuscript! 

Reviewer #2:

Were there eligibility criteria including exclusion or inclusion criteria?

• As we were targeting for a representative population sample there were no specific exclusion or inclusion criteria. Participants were registered at the Qualtrics database and had to be in possession of and able to use a computer. We added this information in the limitations. 

No match between the sample size mentioned in study sample and the result. 

• Thanks for pointing this out, that was formulated somewhat misleadingly. The sample was specified a priori with a minimum of 1000 participants according to age, gender, education, and region. Qualtrics provided us with the final sample of N=1005. All of these participants were analyzed.” 

• Note: Four participants had to be excluded as they were test-participants from Qualtrics. Unfortunately, in our first analysis they were still included due to a misunderstanding between Qualtrics and us. We now recalculated the analysis with the final 1005 participants and corrected all scores throughout the manuscript. The exclusion of the 4 people resulted in no relevant changes in the results.

The sample of your study are people who were in quarantine, and they were asked to fill out 7 questionnaires, what was the response rate? 

• Unfortunately, we do not know how many people were contacted by Qualtrics and therefore cannot report response rate. We highlighted this point in the limitations. 

All 1009 completed the questionnaires without missing one? 

• Correct! The online survey only allowed only to continue by answering all questions (forced choice answer format). As the participants got an expense allowance by completing all questionnaires of € 11,-, there are no missing items in the data set. 

Is this ethical to administer 7 questionnaires? 

• This study was approved by the Ethics Committee of the Danube University Krems (approval code: EK GZ 26/2018-2021) and conducted in accordance with the Declaration of Helsinki. To minimize the duration, we used only short-form questionnaires with mainly five to ten items. 

What was your definition of mental health indicators?

• We wanted to investigate the effect of COVID 19 and relationship on the most prevalent mental health symptoms, such as depression, anxiety, insomnia, or stress, as well as Quality of Life and well-being. We then selected the questionnaires with good psychometric reference values, not too many items, validated in German language and widely used in research.

Please don’t report results in the method like study sample subheading! 

• Thank you for this advice, we removed the mentioned results from the study sample subheading. 

Please don’t compare the result of your study with another as you mentioned in result line 6. 

• Thank you for pointing this out. We removed this comparison. 

Please don’t re-mention the results in discussion. 

• We are grateful for this comment and removed the re-mentioned results from the discussion. 

• Thank you for this constructive feedback and your considerations to improve the quality of our manuscript.

---

## [Editor Report · Decision Letter 1]

27 Aug 2020

Relationship quality and mental health during COVID-19 lockdown

PONE-D-20-15368R1

Dear Dr. Pieh,

We’re pleased to inform you that your manuscript has been judged scientifically suitable for publication and will be formally accepted for publication once it meets all outstanding technical requirements.

Kind regards,

Ali Montazeri

Academic Editor

PLOS ONE

Additional Editor Comments (optional):

1. If you indicate Qualtrics in the Methods would be more informative (Who they are?). 

2. Perhaps if you integrate some explanations in the text (that you have provided for reviewers in response letter) would be better. For instance about missing data or similar. 
---

## [Editor Report · Acceptance letter]

3 Sep 2020

PONE-D-20-15368R1 

Relationship quality and mental health during COVID-19 lockdown 

Dear Dr. Pieh:

I'm pleased to inform you that your manuscript has been deemed suitable for publication in PLOS ONE. Congratulations! Your manuscript is now with our production department. 

Kind regards, 

on behalf of

Professor Ali Montazeri 

Academic Editor

PLOS ONE